# Transcriptomic divergence of network hubs in the prenatal human brain
Stuart Oldham [1,2] & Gareth Ball [1,3] ✉

Connections in the human brain are not uniformly distributed; instead, a dense network of long-range projections converge on highly connected hub regions located in paralimbic and association cortices. Hub connectivity is strongly influenced by genetic factors but the molecular cues guiding the foundation of these structures remain poorly understood. Here, we combined high-resolution diffusion MRI data acquired from 208 term-born neonates with spatially resolved prenatal gene expression data to investigate the molecular correlates of network hub formation at mid-gestation. We identified robust hub architecture in the neonatal connectome and mapped these structural hubs to corresponding cortical regions in the µBrain prenatal digital brain atlas. Transcriptomic analysis revealed differential gene expression in network hubs at mid-gestation, with genes positively associated with hub status supporting the establishment of early neuronal circuitry and predominantly expressed in the transient subplate and intermediate zones. Hub genes were expressed by excitatory neurons, including subplate neurons and intratelencephalic projection neurons in deep cortical layers, overlapped with markers of cortical expansion and interhemispheric connectivity in adulthood and were associated with common neurodevelopmental disorders. Our study identifies prenatal transcriptomic signatures of network hubs in the neonatal human brain, demonstrating how early gene expression programs can shape brain network connectivity from mid-gestation

Higher-order brain functions are founded upon distributed neural activity over anatomical networks[1–4]. Studies of brain network organisation have revealed a set of fundamental properties that span neuronal networks across species and scales[1,5–12]. These include: a hierarchical configuration, a central core of densely connected, high-degree 'hub' regions and, in terms of connection length, near minimal wiring costs[5,10,13–16]. In the human cortex, hub nodes with extensive, distributed connectivity are located in paralimbic and association areas[17–21].

Connections between brain regions are unevenly distributed[22–24]. A disproportionate number of connections form between hubs to forge a central nexus, or 'rich-club'[5,19,25–29]. Compared to non-hub regions, cortical network hubs have higher metabolic energy demands and increased blood flow[30,31]. The high metabolic demand, coupled with dense interconnectivity, of core hubs points to a significant energetic cost of rich-club ordering in brain networks[5,10,27,32]. The cost of a rich network core is balanced by a resilience to perturbations in network connectivity[33] and an increased efficiency of cross-network communication[5,34].

Converging lines of evidence from tracer studies[12,16], human neuroimaging[35–38] and experimental models[39] suggest that the central positioning of hubs within brain networks provides critical infrastructure to integrate information across functional sub-systems. Human fMRI studies show that network hubs activate in concert with segregated, task-specific networks[36–38] and, much like in traffic and information networks, disruption or damage to hub nodes imparts disproportionally widespread disturbances to wider network communication[40–46]. These observations are supported by experimental tracer studies in other mammalian species[4,16,47–49], anatomical lesion studies[41,50–52] and evidence of network hub alterations across a diverse range of psychiatric, neurodevelopmental and neurological brain disorders[50,53–55].

Comparative connectomic studies have revealed that the evolutionary expansion of association cortex in humans is accompanied by a corresponding increase in areal connectivity[56,57]. While the shape of the brain imparts significant constraints on wiring distance[58], computational models suggest that hub locations cannot be explained by cortical geometry or wiring costs alone[59,60]. Instead, hub locations appear grounded in the spatial confluence of genetically programmed molecular gradients during early brain development with spatial differences in developmental timing reflected by commonalities in cytoarchitecture and regional gene expression in core network hubs[18,20,61–63]. Recently, studies have begun to characterise the genetic architecture of white matter connectivity, revealing a highly

[1]Developmental Imaging, Murdoch Children's Research Institute, Melbourne, VIC, Australia. [2]Turner Institute for Brain and Mental Health, Monash University, Melbourne, VIC, Australia. [3]Department of Paediatrics, University of Melbourne, Melbourne, VIC, Australia. ✉e-mail: gareth.ball@mcri.edu.au

polygenic background[20,64,65]. Genetic influences on brain connectivity are concentrated on network hubs and their respective connections, supporting a role of genes in shaping the network core[20]. Together, the vulnerability of network hubs in neuropsychiatric disorders and the strong genetic influence on hub connectivity suggest that the foundation of structural hubs is critical to later brain function[17,66].

Using diffusion MRI, we and others have shown that the structural core of the human connectome is assembled early in development[67,68]. Many fundamental network properties are in place in the human brain by the start of the third trimester[67–70]. The availability of evidence from this developmental period is limited but reveals a right-tailed nodal degree distribution in the neonatal connectome, with high-degree hub nodes located along the cortical midline, insula and in lateral frontal and parietal cortex, forming a core enriched for long-range connections[66–69,71].

In this study, we combine diffusion MRI acquired shortly after birth[72] with a spatial atlas of the prenatal brain transcriptome[73–75] to examine the organising principles of early structural brain networks. By isolating a molecular signature of network connectivity in the prenatal brain, we highlight the role of specific cell populations in the organisation and maintenance of early cortical circuitry and identify putative genetic risk factors for the disruption of developing network structure and brain growth.

## Results

### Highly connected hub nodes in the neonatal cortex

We generated cortical structural connectivity networks using diffusion MRI and whole-brain probabilistic tractography in 208 neonates born at term (101 females; median gestational age at birth [range] = 40 weeks [37- 42$^{+2}$]; median post-menstrual age at scan [range] = 41 [37$^{+3}$–44$^{+5}$] weeks)[72]. Network nodes were defined according to the µBrain cortical atlas, a recently developed 3D neuroimaging-transcriptomic atlas of the developing human

brain[73]. For this study, each cortical region of the µBrain atlas ($n = 29$) was subdivided into similar-sized subparcels of ~90 vertices each ($n = 349$ per hemisphere, mean number of vertices ± S.D = 86.18 ± 14.02), yielding a high-resolution cortical parcellation with boundaries aligned to neonatal cortical anatomy (µBrain$_{90}$; Fig. 1A). Individual tractography streamlines were smoothed[76,77], combined to form a group consensus network and thresholded to include the strongest 15% of edges (Fig. 1B)[20].

Examining the properties of the neonatal connectome, we found that node degree distribution showed a characteristic heavy-tail, indicating the presence of a small population of nodes with disproportionately high degree (Fig. 1C)[2,26]. We calculated the network rich-club coefficient $\phi(k)$ over increasing degree thresholds, $k$, and observed significant rich-club organisation emerging amongst the top 10% of connected nodes ($k > 161$, Fig. 1C). Over all pairs of connected nodes, the median connection length was 53.8 mm, with a higher proportion of long-range connections present between hub nodes (Fig. 1D)[10,27,67,78,79]. Rich-club nodes were distributed across cortical areas, including the insula, cingulate, ventrolateral frontal and dorsal parietal cortex (Supplemental Data 1), mirroring earlier observations in neonatal[67,68] and adult data[20,26] and confirming the early establishment of a structural network core in the human brain (Fig. 1E, F). We further examined rich-club architecture using alternative cortical subparcellations (µBrain$_{60}$; µBrain$_{120}$; Fig. 2A) and network thresholds (5% and 25%) (Fig. S1), observing that estimates of core network connectivity were robust across parameter settings. Further, using repeated subsamples of the full cohort, we found that group-average hub connectivity converged rapidly on the whole-group pattern (Fig. 2B) and remained stable across the perinatal period (Fig. 2C). Repeating this analysis with each participant's structural connectivity data revealed that hub organisation consistent with the group average was present across individual neonatal brain networks (Fig. 2D).

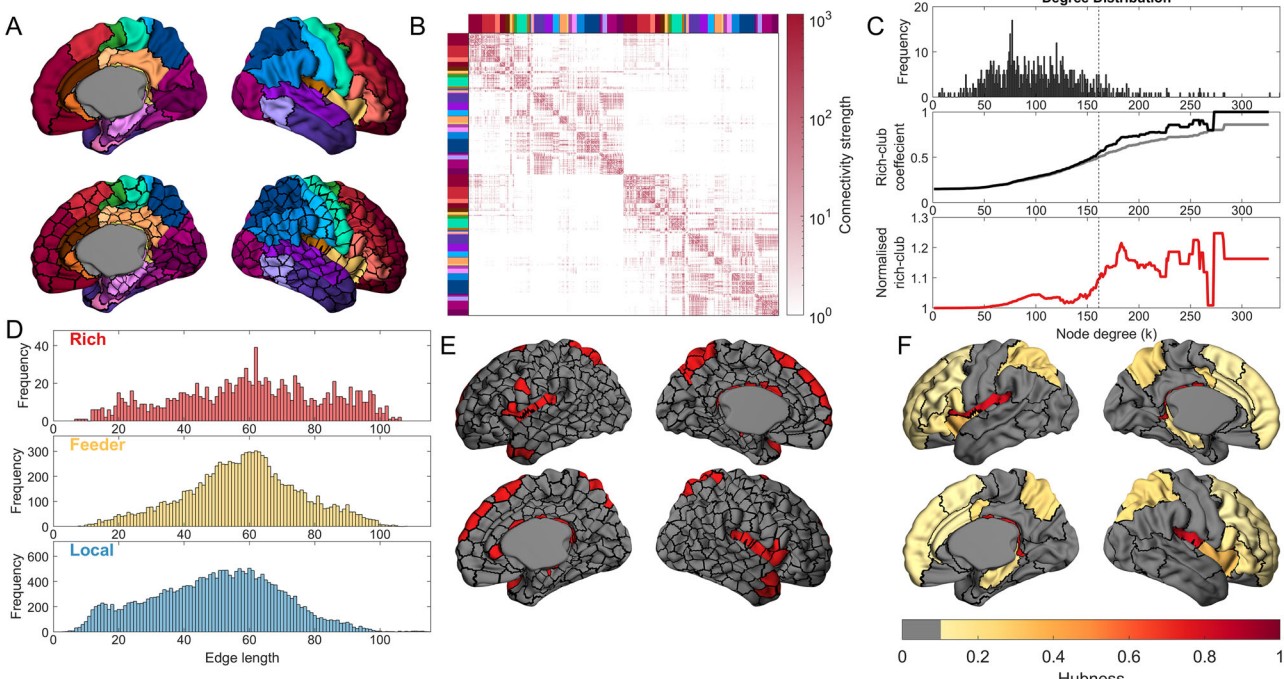

**Fig. 1 | Rich-club hub nodes in the neonatal brain. A** The regions of the µBrain atlas (coloured regions; top) are subdivided into parcels of ~90 vertices (bottom). **B** The group consensus connectivity matrix thresholded to retain the top 15% of edges. Nodes are ordered according to the µBrain region of which they are a subset. **C** The degree distribution (top) of the thresholded, group consensus network. The rich-club coefficient (middle) is calculated over all degree thresholds in the empirical data (black line) and compared to the rich-club coefficient of degree sequence preserving null networks (grey line). Normalised rich-club coefficient (bottom; red line)

values > 1 indicate greater rich-club organisation than expected by chance. The dashed vertical line indicates the 90th percentile for node degree ($k = 161$), above which nodes were considered network hubs. **D** Edge length distributions for rich (connections between hub nodes), feeder (connections between hubs and non-hubs), and local (connections between non-hubs). **E** The location of hub nodes (red) in the neonatal brain. **F** The hubness for µBrain regions, calculated as the proportion of a region's subparcels that were identified as a hub (hubness thresholded at 0.1 for visualisation).

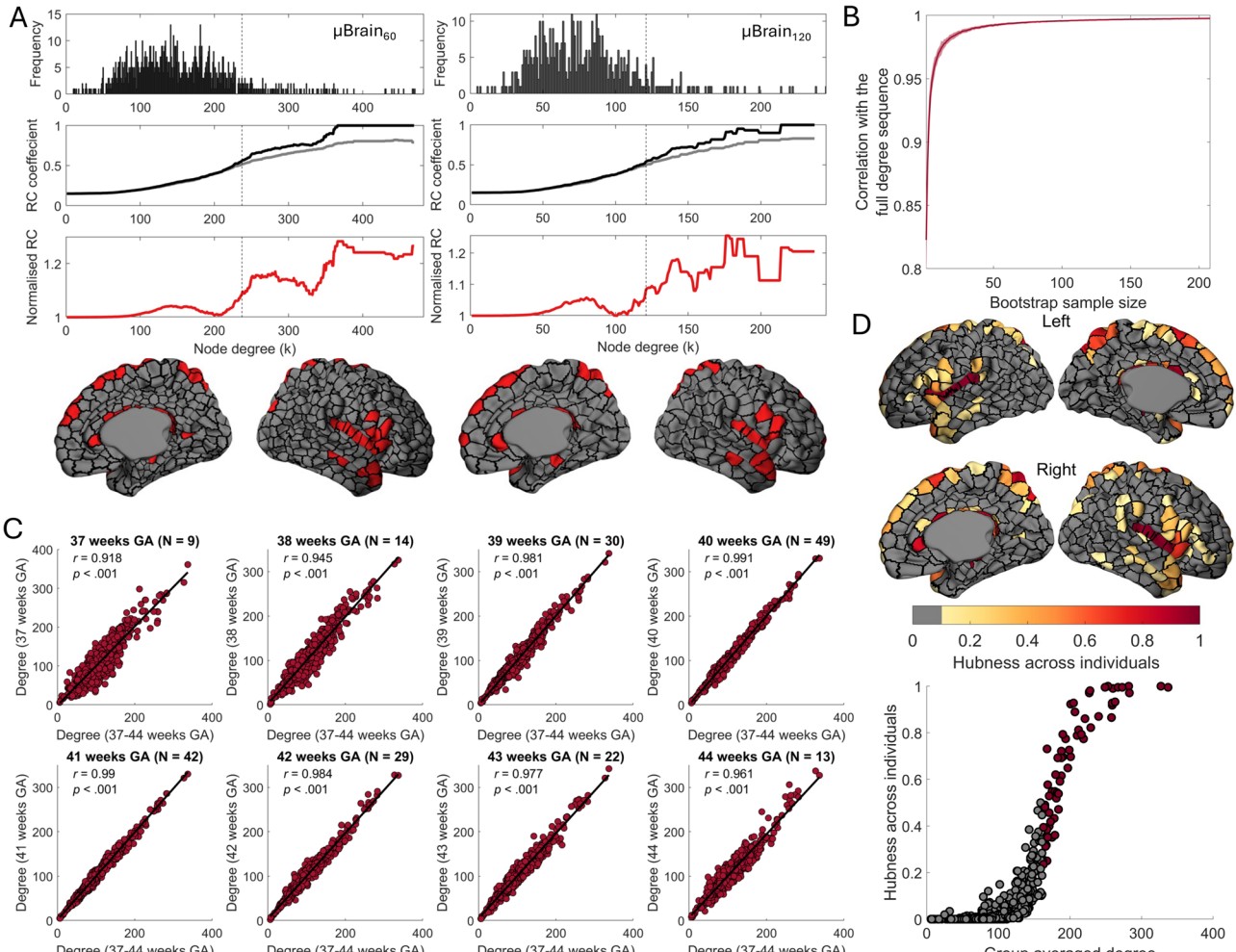

**Fig. 2 | Network hubs are robust across parcellation schemes, ages and individuals. A** Degree distributions (top row) are shown at two alternative resolutions (μBrain60 and μBrain120) with subparcels of ~60 and 120 vertices in area, respectively, with a threshold of 15%. Corresponding rich-club (RC) coefficient curves (middle row) are shown for each as in Fig. 1. Normalised rich-club coefficient (red line) values > 1 indicate greater rich-club organisation than expected by chance. The locations of hub nodes (bottom row) in the left and right hemispheres (coloured regions) are shown overlaid on each parcellation resolution. **B** Correlation between nodal degree profiles in average networks generated from subsamples of increasing size and the whole-group network. The red line indicates the average across the 100

random subsamples, while the shaded red area shows the standard deviation. **C** Correlations between nodal degree of group-averaged networks (μBrain90 thresholded at 15%) created using all individuals scanned within a particular gestational week (ranging from the 37th week to the 44th week) and the whole-group average network. **D** Top, for each individual neonate's structural network, we identified hub nodes (90th percentile for degree). We then calculated the proportion of times a node was identified as a hub across individuals (hubness across individuals). Bottom, individual hubness plotted against the group-averaged degree. Points in red were identified as a hub in the group-averaged consensus network (90th percentile for degree).

## Differential gene expression in neonatal network hubs prior to the time of birth

Cortico-cortical circuitry is established during the second trimester[80–84]. We next sought to test whether the locations of high-degree network hubs are predicated on differences in gene expression in the mid-gestation cortex. Using microarray data from four post-mortem prenatal brain specimens (16–21 weeks post-conception)[74] aligned to the μBrain cortical atlas[73], we tested spatial associations between regional node degree and gene expression independently within five transient tissue zones of the fetal brain.

We identified 644 significant associations (from 574 unique genes; Supplemental Data 2) between prenatal gene expression and the average node degree of each μBrain cortical parcel ($p_{\text{FDR}} < 0.05$, Fig. 3). All significant associations passed additional permutation-based testing to account for potential biases due to spatial autocorrelation in cortical properties ($p_{\text{spatial}} < 0.05$)[85,86]. Most associations were confined to post-mitotic tissue compartments (cortical plate, subplate, intermediate zone; Fig. 3A, B). In total, higher connectivity at birth was associated with increased expression of 289 genes (hub+) and decreased expression of 287

genes (hub−). Similar results were observed across different μBrain parcellation resolutions (Fig. S2). Gene associations were highly correlated across parcellations (μBrain60: μBrain90 $r = 0.996$; μBrain90: μBrain120 $r = 0.989$) with near complete overlap between significant gene sets identified with each resolution (overlap with μBrain90: μBrain60 = 94.9%; μBrain120 = 99.4%).

The hub+ geneset contained several transcription factors including: *CUX1* (subplate: $\beta_{\text{degree}} = 0.0084$, $p = 3 \times 10^{-6}$) and *CUX2* (subventricular: $\beta_{\text{degree}} = 0.0117$, $p = 8 \times 10^{-4}$), factors that regulate dendritic morphology of post-mitotic neurons and proliferation of neuronal precursors in the SVZ[87,88]; *NR4A2* (intermediate zone: $\beta_{\text{degree}} = 0.0196$, $p = 5 \times 10^{-6}$), a specific marker of early-born subplate neurons[89], and *KLF6* (cortical plate: $\beta_{\text{degree}} = 0.0059$, $p = 6 \times 10^{-4}$; subplate; $\beta_{\text{degree}} = 0.0056$, $p = 9 \times 10^{-5}$; intermediate zone: $\beta_{\text{degree}} = 0.0048$, $p = 4 \times 10^{-4}$), which enhances neurite outgrowth in vitro (Supplemental Data 2)[90]. Other hub+ genes that were correlated with network degree included *AMIGO2*, encoding an adhesion molecule involved in dendritic arborisation[91] (cortical plate: $\beta_{\text{degree}} = 0.0124$, $p = 4 \times 10^{-9}$), *EFNA5*, encoding the axonal guidance molecule ephrin-A5[92]

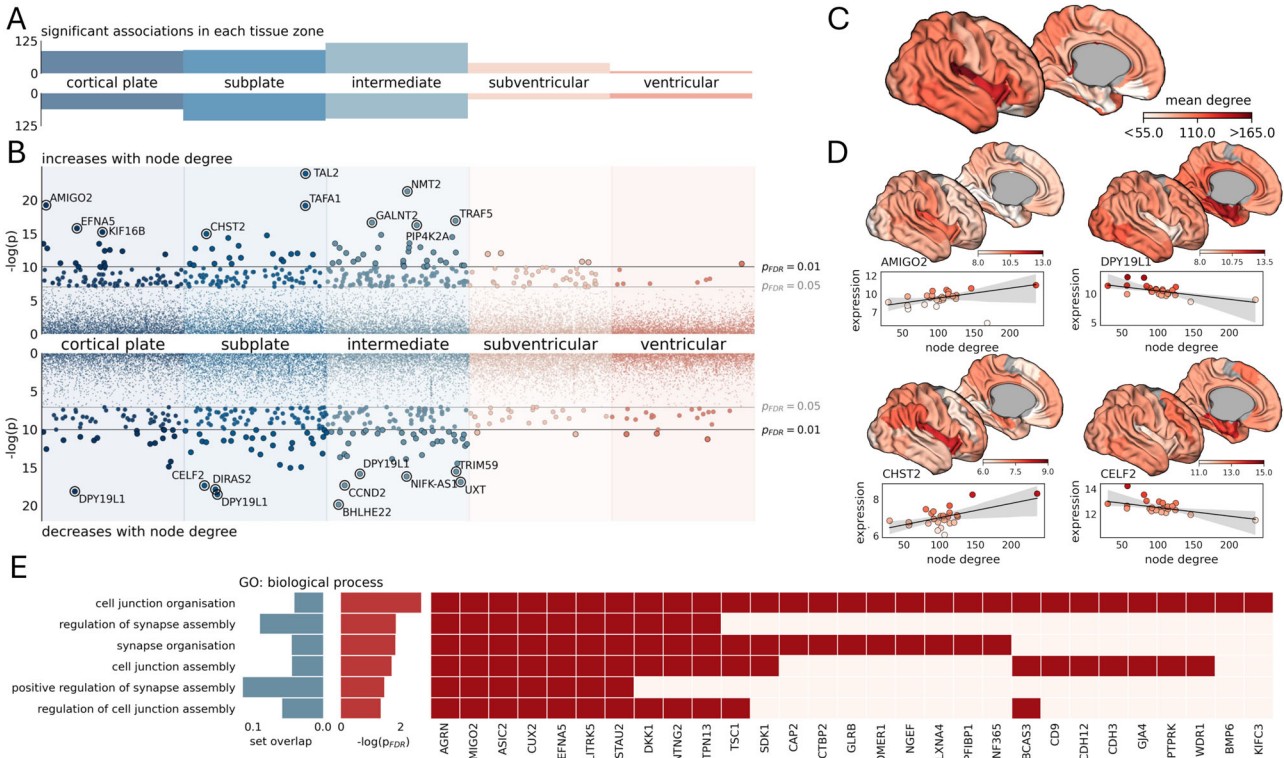

**Fig. 3 | Genes expressed at mid-gestation in cortical regions with high network degree at birth are associated with the development of cortical circuitry.** **A** Number of significant associations ($p_{FDR} < 0.05$) between regional estimates of network degree at term-equivalent age and gene expression in mid-gestation in each of five developmental tissue zones. **B** Individual positive (top) and negative (bottom) associations with node degree for each gene ($n = 7485$ in each zone). The top 10 strongest associations are annotated in each row. **C** Mean node degree of each region in the μBrain cortical atlas. **D** Average (log-normalised) expression of four genes with positive (hub+, left) and negative (hub−, right) associations with node degree, displayed on the cortical surface. Scatterplots illustrate associations between degree and average gene expression, averaged over all brain specimens. **E** Gene ontology (GO) enrichments (FDR-corrected) for biological processes in hub+ genes.

(cortical plate: $\beta_{degree} = 0.0119$, $p = 1 \times 10^{-7}$) and *CHST2*, critical for neuronal plasticity in the developing cortex[93] (cortical plate: $\beta_{degree} = 0.0067$, $p = 1 \times 10^{-4}$; subplate: $\beta_{degree} = 0.0091$, $p = 3 \times 10^{-7}$; Fig. 3D).

Hub− genes included transcription factors involved in neural progenitor fate determination including *EMX2*[94] (intermediate zone: $\beta_{degree} = -0.0104$, $p = 3 \times 10^{-5}$), *MYCL*[95] (subplate: $\beta_{degree} = -0.0050$, $p = 2 \times 10^{-6}$), *SOX11*[96] (subplate: $\beta_{degree} = -0.0036$, $p = 7 \times 10^{-4}$) and its downstream target *NEUROD1*[97,98] (intermediate: $\beta_{degree} = -0.0150$, $p = 9 \times 10^{-7}$), as well as genes required for neuronal differentiation (*CELF2*[99]; subplate: $\beta_{degree} = -0.0065$, $p = 3 \times 10^{-8}$) and radial migration of glutamateric neurons (*DPY19L1*[100]; cortical plate: $\beta_{degree} = -0.0105$, $p = 1 \times 10^{-8}$; subplate: $\beta_{degree} = -0.0130$, $p = 9 \times 10^{-9}$; intermediate zone: $\beta_{degree} = -0.0129$, $p = 1 \times 10^{-7}$; Fig. 3D).

Gene ontology (GO) analysis with FUMA[101] revealed significant enrichment of genes involved in synapse assembly and organisation in the hub+ geneset (Fig. 3E) but no significant GO enrichments in hub− genes.

## Hub genes are expressed by excitatory neurons and astrocytic populations in mid-gestation

Using a comprehensive single-cell atlas of the pre- and postnatal human brain, we examined the expression of hub+ and hub− gene sets across different developmental cell lineages[102]. We found that hub+ were enriched across excitatory neuron lineages in the cortical plate, subplate, and intermediate zones, with specific enrichment in layer 5/6 intratelencephalic (L5/6 IT) and subplate (SP) neurons in the subplate and intermediate zones ($p_{FDR} < 0.05$; Fig. 4A, B; Supplemental Data 3). Hub+ genes in the cortical plate and subplate were also enriched in astrocyte cell lineages ($p < 0.05$, uncorr). In contrast, hub− genes were not significantly enriched in any cell lineage after correction for multiple comparisons (Supplemental Data 3).

Focusing on post-mitotic neuronal and glial populations, we examined the maturational state of cells expressing hub+ genes (Fig. 4C, D). Using pseudotime estimates from single-cell expression profiles[102], we found that total hub+ gene expression increased with maturation in neuronal populations, with the highest expression in differentiated subplate and layer 5/6 neurons. We confirmed that the regional expression of hub+ genes by subplate neurons (hub+$_{SP}$) was spatially correlated to node degree (Fig. 4E, F) and, through examination of independent bulk-tissue RNA-seq data from BrainSpan[101,103], that hub+ genes were enriched during the peak period of subplate development in mid-to-late gestation (Fig. 4G)[104,105]. In glial cell populations, hub+ expression was less well-defined by maturation, with higher levels of expression in both glial progenitors and OPCs, as well as astrocytic cell populations (Fig. 4D).

## Prenatal hub genes are enriched for human cortical expansion and areal connectivity strength in adulthood

Observations from comparative connectomic studies have revealed an evolutionary association between cortical expansion and areal connectivity across mammalian species[56,57,106]. To test for potential shared mechanisms of human cortical wiring and expansion in utero, we compared hub+ and hub− gene sets with genes previously linked to the rate of human fetal cortical expansion[73], revealing significant overlap with hub+ genes expressed in the cortical plate (enrichment = 4.29, $p = 0.007$), subplate (enrichment = 9.80, $p < 0.001$) and intermediate zones (enrichment = 11.48, $p < 0.001$) and with hub− genes in the cortical plate (enrichment = 6.06, $p = 0.002$), subplate (enrichment = 5.78, $p < 0.001$), intermediate (enrichment = 3.76, $p = 0.012$) and subventricular zones (enrichment = 6.28, $p = 0.041$; Supplemental Data 4). Overlapping genes included *CDH12* and *EMID1*, both with roles in cell adhesion and extracellular matrix organisation[107,108], *TRAF5* and *GNAO1*,

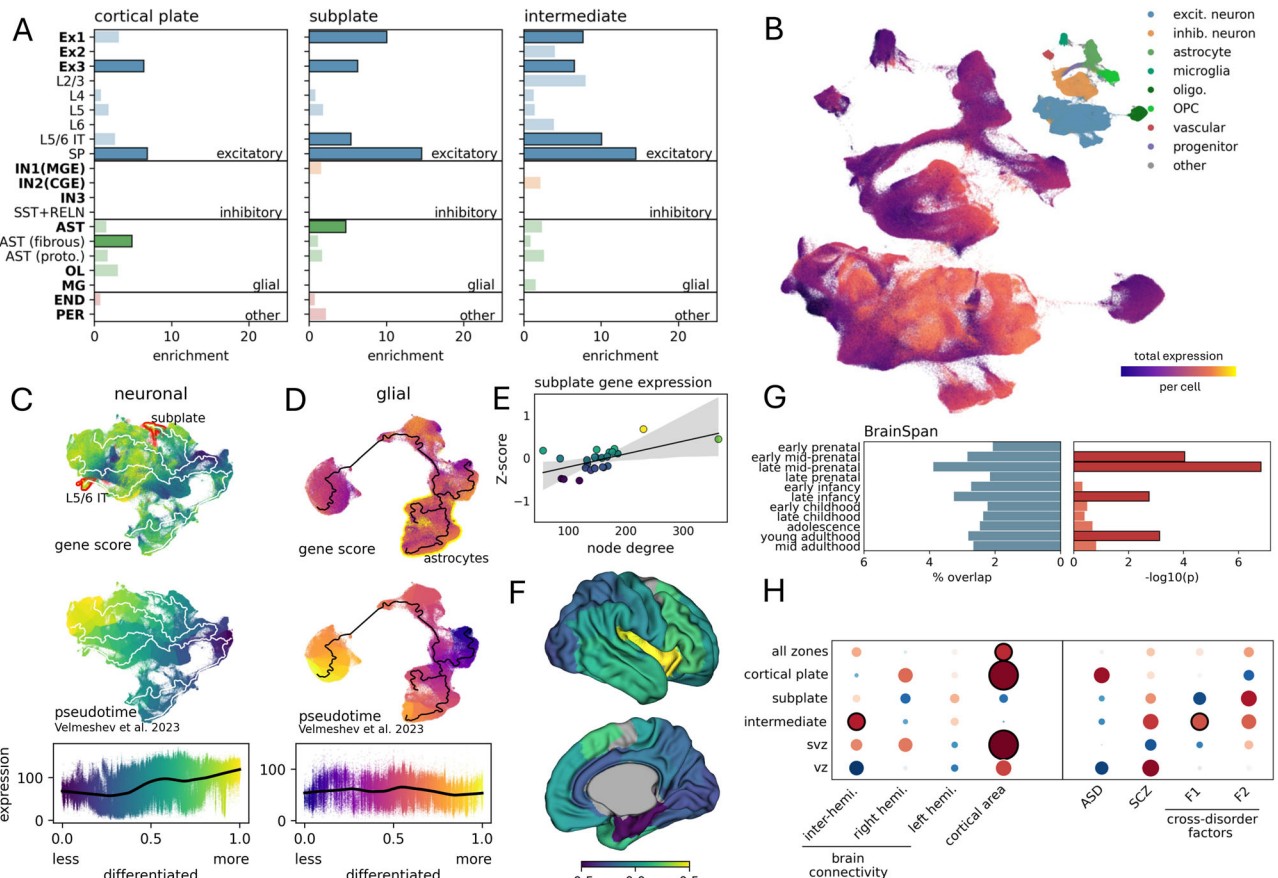

**Fig. 4 | Genes with increased expression in highly connected nodes are enriched for subplate neurons and astrocytes. A** Enrichment of hub+ genes in developmental cell lineages (bold) and cell subtypes in each post-mitotic tissue zone. Significant enrichments (*p* < 0.05 uncorr.) are highlighted with black outlines. **B** Total (log-normalised) expression of network hub+ genes in each of 709,372 cells derived from brain tissue sampled across the human lifespan displayed using UMAP. Inset: territories for major cell classifications. **C** UMAP projections of neuronal and **D** glial cell populations (top), overlaid with developmental pseudotime trajectories. Differentiated subplate neurons, layer 5/6 IT neurons, and astrocyte territories are highlighted with red and yellow, respectively (top row). Plots are coloured by total gene expression of hub+ genes (top) and developmental time (middle). Bottom row:

scatterplots show the relationship between hub+ gene expression and cell maturation for each population. **E** Association between node degree and average expression (Z-scored) of hub+ genes expressed by subplate neurons (hub+$_{SP}$). **F** Average hub+$_{SP}$ expression displayed on the µBrain cortical atlas (lateral and medial surfaces). Additional enrichment analyses are shown in (**G**) and (**H**). **G** Developmental enrichment of hub+ genes across different ages in the BrainSpan RNA-seq dataset. Bold outlines indicate *p* < 0.05. **H** Enrichment in hub+ genes of SNPs associated with different adult brain phenotypes[65,111], neurodevelopmental disorders of early (ASD)[120] and later (SCZ)[121] onset and cross-disorder behavioural factors (F1: mood disorders and F2: psychopathology)[122] (black circles indicate *p* < 0.05 uncorr.).

involved in cell signalling[109,110], and *CUX1*. We also examined associations with corresponding adult brain phenotypes using data from recent GWAS studies[65,111], identifying significant enrichment of loci associated with inter-hemispheric connectivity strength in hub+ genes in the intermediate zone (MAGMA: $\beta = 0.17$, $p = 0.032$) and with cortical surface area in hub+ and hub− genes in the cortical plate ($\beta = 0.49$, $p = 0.001$; $\beta = 0.35$, $p = 0.029$, respectively), intermediate zone (hub−: $\beta = 0.45$, $p = 0.003$) and sub-ventricular zones (hub+: $\beta = 0.49$, $p = 0.018$; Fig. 4H; Supplemental Data 5).

Compared to other primate species, genes differentially expressed in enlarged human cortex (hDEGs) are also enriched in pathways related to synaptic connectivity and are located near to genomic regions with higher substitution rates (human accelerated regions; HARs), deletions (human conserved deletions; hCONDELs) or rapid divergence from primate ancestors (human ancestor quickly evolved regions; HAQERs), suggesting adaptive changes that support increasingly complex brain networks between cortical areas in higher-order primates[56,106,112–115]. We tested if prenatal hub genes were enriched for hDEGs located near human-specialised genomic regions. We found that hub+ genes were enriched for genes located near both HARs and HAQERs (all zones: enrichment = 1.86, $p = 0.0037$; enrichment = 2.93, $p = 0.0004$, respectively; Supplemental Data 4). Tissue-specific enrichments in hub+ gene sets were restricted to the

cortical plate (HARs: enrichment = 2.60, $p = 0.0055$), subplate (HAQERs: enrichment = 5.08, $p = 0.0002$) and intermediate zones (HARs: enrichment = 2.14, $p = 0.015$; hCONDELS: enrichment = 2.37, $p = 0.045$). Hub− genes were additionally enriched for HARs and HAQARS in the cortical plate (HARs: enrichment = 3.30, $p = 0.002$; HAQARS: enrichment = 3.57, $p = 0.027$).

Hub node connectivity is altered across a range of neurodevelopmental disorders (NDDs)[50,54,55,116]. To assess the clinical relevance of hub gene expression in the prenatal brain, we examined the overlap between hub+ and hub− gene sets and single gene mutations linked to ASD (high-confidence *SFARI* genes)[117] and other NDD characterised by brain malformations and/or cognitive sequelae (*Gene2Phenotype* category: definitive)[118]. While several high-confidence NDD genes were identified within hub gene sets, including *AFF2* (Fragile X-E syndrome, hub+), *GNAO1* (epileptic encephalopathy, hub+), *SMARCA2* (Nicolaides-Baraitser syndrome, hub+) and *RTTN* (polymicrogyria, hub−), we found no significant enrichment of pathogenic ASD or NDD variants in either hub+ or hub− genes outside of the ventricular zone, where 2 out of 8 hub+ genes (*TSC1*, *ANP32A*; enrichment = 15.78, $p = 0.007$) were identified (Supplemental Data 4). We performed additional enrichment analyses using MAGMA[119] across an array of previously published genome-wide association studies

(GWAS; Fig. 4H). We observed a weak enrichment of SNPs associated with ASD[120] in prenatal hub+ genes expressed in the cortical plate ($\beta = 0.169$, $p = 0.057$; Fig. 4H; Supplemental Data 5), whereas hub− in the cortical plate were enriched for schizophrenia[121] ($\beta = 0.263$, $p = 0.042$). Additional weak associations were observed for hub+ genes in the intermediate zone for transdiagnostic symptoms of mood disturbance (F1; Fig. 4H; $\beta = 0.162$, $p = 0.049$) and psychopathology (F2; $\beta = 0.149$, $p = 0.079$)[122].

## Discussion

Cortico-cortical networks are critical for brain function[1–5]. The foundations of macroscale brain networks: richly-connected hubs that support stereo-typed patterns of connectivity between distributed cortical areas, are in place by the time of normal birth[26,67,68]. In this study, we uncover a molecular signature of core network hubs captured in mid-gestation and enriched for developmental processes critical for the establishment and organisation of interareal circuitry.

Hub locations in the human brain network are conserved across development[123] and consistent with those borne out by experimental tracer studies in non-human primates and other mammalian species[25,29,124]. During brain development, the differentiation of cortical areas is guided by morphogenic transcription factors along a spatio-temporal schema established early in gestation[125–127]. Recent neuroimaging evidence has shown that the geometry of the cortex plays a significant role in the organisation of network connectivity, with structural and functional properties of the mature cortex varying continuously across the cortical sheet, reflecting early spatial gradients[58,128,129]. Subsequently, the areal patterning of cytoarchitecture, axonal connectivity, and function is undepinned by spatial variations in gene transcription across the cortex[20,125,130,131]. Regional connectivity is both highly heritable and tightly coupled to cytoarchitectural and transcriptomic similarity in the adult brain[20,65,130] suggesting the locations of hub regions in cortical networks are constrained by genetic factors. We find that at mid-gestation, putative hub regions were associated with increased expression of genes supporting neuronal growth, synaptic plasticity, and circuit formation compared to non-hub regions, which were characterised by genes linked to earlier progenitor and migratory processes (e.g., neural progenitor fate determination, radial migration). This pattern signals an accelerated developmental timeline in cortical hubs in place by mid-gestation, potentially allowing for an extended period of circuit formation. At the cellular level, areal differences in gene expression are evident by 20 weeks of gestation[74,132,133] with a recent spatial transcriptomic study identifying somewhat opposing modes of arealisation in the cortical plate in mid-gestation, observing gradual transitions along the anterior-posterior axis overlaid by sharp transitions between similar cell types in anatomically adjacent areas[134]. Whether early network hubs emerge at the confluence of broader organisational gradients, or at locations with tightly defined areal boundaries, remains to be determined.

Neurogenesis does not terminate across the cortex simultaneously, ending first in (para)limbic and allocortical structures before following a broadly rostral-caudal axis across the neocortex[135–139]. Computational modelling reveals that this form of areal heterochrony can produce networks with similar properties to empirical brain networks, including high-degree hub nodes[63]. Thus, early completion of neurogenesis may afford hub nodes a prolonged period of circuit integration, following a 'rich-get-richer' wiring principle[140]. Indeed, many hub regions display decreased neuronal density[62,141], a weak eulaminate structure[142], and denser dendritic arborisation[143], cytoarchitectural properties that imply the earlier completion of neurogenesis[136,137,141,142]. As an example, the insula, identified here as a key network hub in the neonatal brain, is one of the first cortical areas on the lateral surface to mature[144]. Though the insula lacks its own proliferative zone, neuron migration from the pallial-subpallial border occurs early in gestation[145], and by 20 weeks, synaptic density is higher in the insula than in other cortical areas[146]. Consequently, delta brushes—electrophysiological hallmarks of the maturing cortex—arise first from the insula, at around 30 weeks gestation[147].

The early formation of hub circuitry may confer an additional advantage due to the high wiring cost associated with longer cortical connections. In line with earlier reports, we find that connections between hub nodes are, on average, longer than those between non-hubs[5,67]. During brain development, exuberant interareal outgrowth of axonal connections is followed by a period of refinement, with extraneous connections pruned to define the mature connectome[130–132]. A head start in forming connections, taking advantage of the compact size of the human brain earlier in gestation, would reduce the high metabolic costs of establishing critical long-range connections through non-specific exuberant growth[17,66,79,140]. The enrichment of hub genes in human-specialised regions of the genome further supports evidence of adaptive mechanisms in place that may mitigate the cost of long-range network connectivity in the expanded human brain[18,56].

Due to the inherent challenges of acquiring high-quality fetal diffusion MRI data[148–151], few studies have examined the emergence of structural brain network properties *in utero*[152], with most studies relying on the *ex utero* examination of preterm infants[67–69]. In a recent example, Chen et al. used diffusion MRI to define cortico-cortical networks from 26 weeks, revealing a significant increase in connectivity over the second and third trimester in putative hubs, including the cingulate and superior parietal cortex[152]. However, due to acknowledged difficulties in fetal acquisition and processing, interhemispheric connections were excluded from the analysis, yielding an, as yet, incomplete picture[152]. Focusing on individual tracts, post-mortem anatomical studies have demonstrated that the path of major commissural and projection fibres can be traced from mid-gestation with diffusion MRI[83,84,153], with similar results recently reported in vivo[150]. But, while the organisation of immature axons into major white matter bundles is evident from as early as 10 weeks, synapses do not form in the developing cortical plate until after 20 weeks[154,155] with axon terminals accumulating first in the transient subplate 'waiting zone'[80,104,105].

Subplate neurons are among the earliest born and maturing cells of the cerebral cortex[89,156], settling subjacent to the cortical plate to facilitate the emergence of neuronal circuitry and assist with the guidance of intratelencephalic and thalamocortical axons to their final cortical targets[80,104,105,157]. Here, we find that genes expressed early in densely connected hub nodes are enriched in differentiated subplate neurons, including the canonical marker *NR4A2*[89], as well as early differentiating, intratelencephic deep layer (L5/6) neurons. Similarly, we identified a moderate enrichment of hub+ genes in astrocytes, key participants in early neural circuit formation[158]. Subplate neurons are critical to the establishment of thalamocortical connections[104], with their removal preventing the functional maturation of the cerebral cortex[159]. As well as their prominent role guiding thalamic innervation, subplate neurons also connect to each other over long distances, even across hemispheres[160–163], thus providing an initial substrate for cortico-cortical communication. Our findings suggest that, from as early as 15 weeks of gestation, putative cortical hubs can be characterised by the transcriptional signature of circuit formation and synaptic assembly in the developmental subplate. In humans, the subplate in association areas is thicker and persists longer into gestation compared to other regions[156,164]. This elongated period of developmental circuit formation may be vital to the increased complexity and capacity of hub areas in human brain networks[165].

We identified several hub genes with pathogenic variants linked to NDDs, including *AFF2, GNAO1,* and *RTTN,* but only weak enrichment of GWAS loci associated with ASD and SCZ. Hub dysconnectivity has been identified as a potential substrate for a number of brain disorders, suggesting a potential developmental vulnerability that may result in abnormal neuronal circuitry[50,53–55]. Although these conditions are thought to partially emerge from atypical developmental processes, the lack of strong genetic associations in our findings may reflect that our data is only a snapshot of the development of network connectivity in the fetal brain, focused on the establishment of neural circuitry in mid-gestation. Indeed, other factors are clearly involved in shaping early network connections. One of the most notable is the innervation of thalamocortical connections. Thalamocortical connections have a key role in shaping the arealisation, connectivity, and cytoarchitecture of the

cortex[105,125,126,137,166]. To gain a comprehensive view of how genetics shape brain network organisation, the contributions of non-cortical regions should be considered by future work. We also acknowledge that our observations are limited by the relatively short time window of the microarray data currently available for analysis. While our findings align with other evidence that hub areas follow a distinct developmental trajectory that begins earlier than in non-hub regions[17,66,140], gene expression varies markedly during development[103,133]. Examination of the concurrent changes in regional gene expression and cortical connectivity over gestation would require both i) improved spatial and temporal sampling of prenatal gene expression data and ii) reliable and robust fetal diffusion MRI acquisition and tractography protocols. Unfortunately, despite a growing catalogue of comprehensive single-cell RNA-seq studies, including those that span the whole of gestation[102,167], there are not, as yet, datasets of prenatal gene expression that encompass mid- to late gestation at a spatial resolution comparable to the microarray data used here. A more comprehensive evaluation of the temporal development of hub gene expression during this critical window is warranted when such data is available. Similarly, due to difficulties in image reconstruction, the μBrain atlas doesn't contain labels for the outermost cortical layers: the marginal and subpial zones, thus our analysis does not consider the potential role in the development of cortical connections of early-born cell populations in these zones[168].

A further limitation of our work is that we based our assessment of hub connectivity on binary network topology. While the number of axonal connections between regions spans orders of magnitude[49,169], appropriately defining weights such that they reflect the underlying strength of axonal connectivity using diffusion weighted imaging metrics remains challenging[170,171]. However, in brain networks, weighted and unweighted measures of hubness tend to coincide[172], and we are confident that binary topology is sufficient to identify the key network hubs in the developing brain. Incorporation of multimodal data, in the form of morphometric networks, or structural-functional associations will likely provide additional insight into the development of brain connectivity in future studies[173–175].

Taken together, our findings reveal that the establishment of hub connectivity, vital for long-range integration across brain networks, is underway by mid-gestation and characterised by distinct, spatially patterned programmes of gene expression *in utero*.

## Methods and materials
### Participants
Participant data were acquired from the third release of the Developing Human Connectome Project (dHCP)[72]. Ethics approval was granted by the United Kingdom Health Research Ethics Authority, reference no. 14/LO/1169. The full cohort comprised 783 neonates (360 female; median birth age [range] = $39 + ^2$ weeks $[23–43 + ^4]$) across 887 scans (median scan age [range] = $40 + ^6 [26 + ^5 –45 + ^1]$ weeks; 104 neonates were scanned multiple times). For this study, only neonatal scans acquired from term-born infants with a radiological score of 1 or 2 (indicating no/minimal radiological abnormalities or pathologies) that also met additional quality control for in-scanner motion criteria (see below) where included, resulting in a final cohort of 208 neonates (101 females; median gestational age at birth [range] = $40^{+1}$ weeks $[37–42^{+2}]$; median post-menstrual age at scan [range] = 41 $[37^{+3}–44^{+5}]$ weeks).

### MRI acquisition and processing
Images were acquired on a Phillips Achieva 3 T scanner at St Thomas Hospital, London, United Kingdom, using a dedicated neonatal imaging system[72,176]. T2-weighted Fast Spin Echo (FSE) multislice images were acquired in sagittal and axial orientations with overlapping slices (TR = 12,000 ms; TE = 156 ms; resolution = 0.8 × 0.8 × 1.6 mm, 0.8 mm overlap). Sagittal and axial image stacks were motion corrected and reconstructed into a single 3D volume[177]. Diffusion MRI was acquired

with a spherically optimised set of directions over 4 b-shells (20 volumes × b = 0 s/mm²; 64 directions × b = 400; 88 × b = 1000; 128 × b = 2600)[178,179] with a multiband factor acceleration of 4, TR = 3800 ms; TE = 90 ms; SENSE: 1.2 and acquired resolution of 1.5 mm × 1.5 mm with 3-mm slices (1.5-mm overlap) reconstructed using an extended SENSE technique into 1.5 × 1.5 × 1.5 mm volumes[180,181].

Structural images were processed using the dHCP's minimal pre-processing pipeline, including bias correction, brain extraction, tissue segmentation, and cortical surface reconstruction[182]. Diffusion data were processed using a neonatal-specific pipeline[183]. This included correction of susceptibility and eddy current-induced distortions, motion artefacts, and signal dropout[184–187], automated extraction of quality control (QC) metrics, and nonlinear alignment to the 40-week dHCP neonatal template via each subject's corresponding anatomical data[183,188–191]

### In-scanner motion quality control
To ensure only high-quality scans without major motion-related artefacts were used, we examined the quality control summaries of the dHCP diffusion processing pipeline[183]. Scans more than two standard deviations away from the mean on any of the volume-to-volume motion, within-volume motion, susceptibility-induced distortions, and eddy current-induced distortions metrics were excluded from further analysis.

### Connectome reconstruction
Following pre-processing and QC, each subject's diffusion data was used to generate whole-brain tractography following a previously developed processing pipeline in MRtrix3 (v3.0.2)[77,192]. As in our previous work[77], we modelled the diffusion signal using only the 0 and 1000 s/mm² b-value diffusion volumes. This approach has been shown to provide a better definition of FODs in neonatal data compared to multi-shell approaches[193]. We estimated a white matter response function from the oldest 20-term neonatal scans using the *dhollander* algorithm[194,195]. Using this neonatal response function, fibre orientation distributions were calculated for each participant using single-shell 3-tissue constrained spherical deconvolution (CSD)[195]. Probabilistic whole-brain tractography was performed using second-order integration over fibre orientation distributions (iFOD2)[192] (0.75 mm step size; 45° maximum angle; 0.05 fibre orientation distribution cutoff), with Anatomically onstrained ractography[196].

The parcels of the μBrain parcellation vary substantially in size, thus to limit potential bias due to large variations in surface area between nodes[197], we divided each region of the μBrain parcellation into smaller subparcels with an approximately equal number of vertices (n = 90), yielding the μBrain₉₀ parcellation with a total of n = 698 cortical subparcels. To test the impact of network resolution, we created additional divisions of subparcels with an average of 60 (μBrain₆₀; 1032 subparcels) and 120 (μBrain₁₂₀; 528 subparcels) vertices.

To generate brain networks from the whole-brain tractograms, we first applied connectome spatial smoothing (CSS)[76,77]. CSS smooths and streamlines counts across vertices of a cortical mesh to improve the reliability and robustness of individual whole-brain tractograms[76]. To create a cortical mesh for CSS, the dHCP neonatal 40-week white matter surface was aligned to each individual's diffusion space using transforms provided by the dHCP. Streamlines were assigned to the nearest cortical vertex of this mesh within a 5 mm radius of their endpoint, with a Gaussian smoothing kernel applied (3 mm FWHM, 0.01 epsilon) to adjust the strength of connectivity across cortical vertices, creating a high-resolution connectome. The high-resolution connectome was mapped to the μBrain₉₀ parcellation by summing smoothed streamline counts over all vertices within each subparcel.

### Hub definition
Hubs were defined as previously described in adult data[20]. We initially constructed a group consensus network by selecting connections that were (i) present in over 30% of individual networks, and (ii) in the top 15% of connections by strength.

To identify hubs within the consensus network, the rich-club coefficient was calculated across $k$ degree thresholds as:

$$\phi(k) = \frac{2E_{>k}}{N_{>k}(N_{>k}-1)}$$

where $N_{>k}$ is the number of nodes with degree $> k$, and $E_{>k}$ is the number of edges between nodes with degree $> k$. As nodes with a higher degree are more likely to be connected to each other by chance, we generated 100 random networks and computed the rich-club coefficient $\phi_{rand}(k)$ for each of these networks. The random networks were created by rewiring each edge of the group consensus network 50 times while retaining the degree sequence of the original network. The normalised rich-club coefficients were calculated as the ratio between the group consensus rich-club coefficient and the mean rich-club coefficient across the random networks for a given degree threshold, $k$:

$$\phi_{norm}(k) = \frac{\phi(k)}{\langle \phi_{rand}(k) \rangle}$$

Values of $\phi_{norm}(k) > 1$ indicate that high-degree nodes are more densely interconnected than would be expected by chance, revealing the presence of rich-club organisation. Statistical significance is evaluated by calculating a $p$ value using the empirical null distribution of $\phi_{rand}(k)$, derived from 100 randomised networks.

Inspection of the rich-club coefficient curves indicated that high-degree nodes form a rich-club regime beginning at $k > 161$. Therefore, we designated all nodes with degree $> 161$ as hubs. To map hubs to the original μBrain parcellation, we generated an areal 'hubness' index defined by the proportion of subparcels within a given μBrain region designated as hubs, termed $RC_{\%}$.

### Transcriptomic data processing

Prenatal microarray data were made available as part of the BrainSpan database [https://www.brainspan.org/]. For details on tissue processing and dissection, see Miller et al.[74] Normalised microarray data were obtained from 1206 tissue samples across the left hemisphere of four post-mortem fetal brain specimens (age 15–21 PCW, 3 female)[74]. As described in prior work, each tissue sample location was matched to corresponding regional labels and tissue zones (cortical plate, subplate, intermediate zone, subventricular zone, ventricular zone) as part of the μBrain atlas[73]. Samples that were not matched to labelled regions, including samples from subcortical nuclei, midbrain structures, subpial granular and marginal layers, and brainstem, were removed. Low signal probes designated 'absent' were also removed (34.67% of probes). Where multiple probes mapped to a single gene, the probe with the highest differential stability (DS) was assigned[198]. We calculated DS as the average pairwise correlation of sample expression within each tissue zone and between pairs of specimens sampled at the same time point. Probes with DS < 0.3 were removed.

Where more than one sample was available for a given region or zone, e.g., samples from the outer and inner cortical plate in the same region, gene expression was averaged across samples. Finally, any probes with missing data in more than 10% of tissue samples were removed ($n = 2271$). This resulted in expression data from 7457 genes across 27 regions and 5 tissue zones for analysis.

### Single-nucleus RNA data

Harmonised and log-normalised single-nucleus RNA sequencing profiles for 709,372 cells were downloaded from the UCSC Cell Browser (cells.ucsc.edu/?ds=pre-postnatal-cortex)[102]. Cells were sampled from 106 post-mortem brain tissue samples aged from approximately 16 gestational weeks to 54 years. Additional data included cell lineage gene markers, UMAP projections and pseudotime trajectories. For full details, refer to Velmeshev et al.[102]

### Statistical analysis

For each gene, we used a general linear model (GLM) to test the hypothesis that mid-gestation gene expression was associated with node degree (total number of connections) in each cortical region. GLMs were performed for each gene in each of 5 tissue zones (total number of tests = 7457 per zone), including the age of each specimen (15/16 or 21 PCW) as a covariate. Robust linear models were fit to expression data using iteratively reweighted least squares to account for potential outliers. Significant associations between expression and node degree were identified after multiple comparison correction for False Discovery Rate within each tissue zone ($p_{FDR} < 0.05$).

To account for the potential confounding effects of spatial auto-correlation inflating associations between hub location and gene expression[85,86], we generated 1000 surrogate maps with spatial auto-correlation and areal weights matched to the group-average node degree maps using the software tool, *BrainSMASH*[86]. For each gene association, we re-fit the statistical model to each surrogate map to estimate a null distribution of coefficients. Significance of the observed value was based on a $p_{spatial} < 0.05$ under the null distribution.

### Enrichment analyses

GO enrichment was performed using *gene2func* implemented in FUMA[101]. SNP enrichment was performed using MAGMA[119] and summary statistics from recent GWAS studies[65,111,120–122]. For other enrichment analyses, we calculated enrichment as the ratio of the proportion of genes of interest and the proportion of background genes within each gene set. Unless otherwise stated, the background set was defined as the full list of genes included in the study ($n = 7457$). Significance was determined using the hypergeometric statistic:

$$p = 1 - \sum_{i=0}^{x} \frac{\binom{K}{i}\binom{M-K}{N-i}}{\binom{M}{N}}$$

where $p$ is the probability of finding $x$ or more genes from a specific geneset $K$ in a set of randomly selected genes, $N$ drawn from a background set, $M$.

### Reporting summary

Further information on research design is available in the Nature Portfolio Reporting Summary linked to this article.

### Data availability

Neuroimaging data for the Developing Human Connectome Project are available via the NIMH Data Archive (collection ID: 3955). The μBrain atlas and associated data are available at: https://zenodo.org/records/10622337. Preprocessed single-nucleus RNA-seq data used in this study were downloaded from: http://cells.ucsc.edu/?ds=pre-postnatal-cortex and associated publications. Source data underlying the graphs and charts presented in the main figures are available in Supplemental Data 6.

### Code availability

Supporting code for this manuscript is available at: https://github.com/garedaba/prenatal-hubs

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

## Acknowledgements

This research was supported by the Brain and Behaviour Research Foundation (31471 to S.O.), an NHMRC Investigator Grant (1194497 to G.B.), Murdoch Children's Research Institute, The Royal Children's Hospital, and the Victorian Government's Operational Infrastructure Support Programme. The project was generously supported by The Royal Children's Hospital Foundation, devoted to raising funds for research at The Royal Children's Hospital.

## Author contributions

Conceptualisation, methodology, software, analysis, writing, revision: S.O. & G.B. supervision, resources: G.B.

## Competing interests

The authors declare no competing interests.
