## [Transparent Peer Review file · Communications Biology]

Transcriptomic divergence of network hubs in the prenatal human brain

Corresponding Author: Dr Gareth Ball

Version 0:

Reviewer comments:

Reviewer #1

(Remarks to the Author)

This study represents a highly novel and instructive investigation characterizing the transcriptomic underpinnings of neonatal brain connectome. Using diffusion MRI data from the dHCP project, the authors identified hub regions of the brain white matter network at birth and further integrated multiple existing brain transcriptomic atlases (uBrain, BrainSpan, Single-cell genomics atlas) to uncover a system and cross-scale molecular blueprint on the origin of cortical hub. They found that the neonatal hub pattern is strongly associated with gene expression in the transient subplate and intermediate layers during mid-gestation, particularly involving genes related to dendritic morphology and axonal guidance, and predominantly expressed by excitatory neurons and astrocytes. The authors are domain experts in neuroimaging-transcriptomic analysis. For connectome analysis, they effectively addressed common pitfalls such as node parcellation, fiber smoothing or threshold selection. For genetic analysis, the uBrain atlas represents as a methodological contribution from their prior work. Overall, this study leverages rare data and rigorous methods to provide novel insights into the developmental timing and genetic regulation of canonical white matter network architecture. It is encouraging to see these findings, and only a few suggestions remain that do not affect the manuscript's suitability for publication.

Major suggestion:

1) The primary challenge in identifying the transcriptomic basis of neonatal hub regions lies in establishing a reliable hub distribution. The authors have conducted solid validations regarding parcellation schemes, network thresholds, and individual similarity. However, I have suggestions for further strengthening.

a) Considering the rapid changes of brain at early stage, is there any development effect of hub location shortly after birth? If such changes are significant, should they be incorporated into the gene-level analyses? b) the convergence of group-level nodal degree profiles with increasing sample size could be evaluated (is a sample size of 200+ sufficient?); c) Figure S4 shows consistent associations between nodal degree and gene expression across three network resolutions; however, the reproducibility of specific genes across resolutions is not reported. If the identified gene sets are largely overlapping, reporting this would largely strengthen the robustness; if not, only focusing on the resolution-robust gene subsets may be a good choice. d) some of the validation results in the supplemental figures could be integrated into Figure 1 to better illustrate the stability of hub distribution.

Minor suggestions:

1) In the intro part, they mentioned a few times of the modular structure of human brain connectome, which seems unrelated to the main focus. I recommend removing them.

2) The sentence "The central positioning of hubs within brain networks supports integration across functional sub-systems; a framework that is supported by experimental tracer studies in other mammalian species" conveys a crucial point. Could the authors elaborate on how these hubs support specific functional subsystems?

3) The sentence "However, computational simulations predict that disruptions to brain hubs and their connections have far-reaching and adverse effects on overall network function" seems lack a clear contrast with the preceding context. Please make a further clarification.

4) In addition to multiple comparison correction (FDR), null model corrections (e.g., spin tests, random rewiring) are also critical in gene-hub association analyses and may need to be considered.

5) The uBrain atlas only covers up to 21 gestational weeks, while the neonatal brain scans were acquired at a mean age of around 40 weeks. Given the substantial brain changes during this period, particularly the disappearance of the subplate zone implicated in the current findings, it would be helpful to further discuss this temporal gap.

- 6) The neonatal dMRI scan contains multi-shell DWI volumes, however, the authors seems only employed single-shell models for diffusion modeling. Please clarify the rationale.
- 7) It may be helpful to split Figure 3 into two figures according to the current result sections, which could improve readability.

Reviewer #2

(Remarks to the Author)

The manuscript Transcriptomic divergence of network hubs in the prenatal human brain is an ambitious attempt to combine two sets of data (MR and transcriptomic atlas), two developmental periods (mid-gestation and newborn), in order to get new data on development of the cortical network hubs in the developing prenatal human brain. According to authors, this concept can be called "binary topology". The subject is very important because data on developing networks during first half of gestation are insufficient, but very important for analysis of major developmental disorders, especially schizophrenia. The abridgements of these two sets of secondary data was performed by careful statistical evaluation of relationship between newborn connectomic data, parcellation of telencephalic pallium and genes expressed at mid-gestation in cortical regions with high network degree at birth. The data are very well documented, illustrated and adequately complemented in supplemental material. The paper offers original conclusions about role of early compartmentalization of transcription factors, claiming that a head start in the formation of the connectivity reduces the high metabolic cost of establishing cortical long-range connections. The data also fill the gap in current neuroimaging studies, where dynamically changing transient anatomical compartments and specific neural and glial classes are not considered. In respect to this last question, they found that excitatory projection neurons in deep layers, such as subplate and layer 6, show specific spatially resolved prenatal gene expression, which are important for establishment of network hubs prior to the time of birth. Conceptually, I have some difficulties in interpreting tight association between transcriptomic divergence in mid-gestation and MR connectomic data in newborn. First of all, compartments in midgestational brain are actually tangentially oriented from pole to pole and from limb to basis of the brain. While arealization is event of second half of gestation and I imagine that hubs are focal points of unknown termination and unknown origin of pathways. I suggest that this problem tangential vs hub discuss with somewhat more elaborative approach. The data on development of connectivity are very important for different disciplines and in this respect paper may have significant impact on broader audience.

Specific points:

Title: The title does not give indication on extensive newborn MR data. Maybe this can be added in title. Description of Figure 2 can serve as a model.

Abstract. Row 41. Transmodal is also association cortex. Row 47. Prior to time of birth. It is not precise enough. Can you mention significance for study of neurodevelopmental disorders?

Introduction. Row 69-72. Expression that anatomical lesions correlate with hubs is too strong, but this is a good idea to do in the future. Row 85. By 30 weeks of gestation pathways all wiring is in position. This is actually the last trimester. Row 76-79. Excellent compression of current evidence, but maybe difficult for non-specialist to reveal these complex relationships. Row 84-94. Despite advances of MR analysis of connectome there is no agreement how many parcels we have. For example, classical neuroanatomist Rose described almost hundred areas in entorhinal cortex.

Results. Row 134. Be more precise, across radial or tangential plane? Row 137. Add postmigratory compartments. Mention that marginal zone cannot be visualized and it is difficult to perform transcriptomic analysis. Row 234. Because of these results few words on neurodevelopmental disorders should be included in Abstract.

Discussion. I would emphasize significance of binary and multimodal approach for association between hubs and transcriptomics of transient compartments. It is important to elaborate so-called limitations of the study from Row 309 to 320. Please consider prominent changes which occur between mid-gestation and newborn age. For example, associative neurons of layer 3, which send long connections are not even born before 13 weeks post conception. Row 287. When quoting about origin of subplate consider that only papers by Rakic and Kostovic group give data on origin of subplate in human and monkey (see A. Duque, Z. Krsnik, I. Kostović, & P. Rakic, Secondary expansion of the transient subplate zone in the developing cerebrum of human and nonhuman primates, *Proc. Natl. Acad. Sci. U.S.A.* 113 (35) 9892-9897, <https://doi.org/10.1073/pnas.1610078113> (2016).

References, I recommend two references: one about multimodal gradients of cortical organization and the other, because of frequent mentioning of synapses, recent paper on multiscale analysis of synaptic distribution:

Wang Y, Eichert N, Paquola C, Rodriguez-Cruces R, DeKraker J, Royer J, Cabalo DG, Auer H, Ngo A, Leppert IR, Tardif CL, Rudko DA, Leech R, Amunts K, Valk SL, Smallwood J, Evans AC, Bernhardt BC. Multimodal gradients unify local and global cortical organization. *Nat Commun.* 2025 Apr 25;16(1):3911. doi: 10.1038/s41467-025-59177-4. PMID: 40280959; PMCID: PMC12032020.

Kostović I. Development of the basic architecture of neocortical circuitry in the human fetus as revealed by the coupling spatiotemporal pattern of synaptogenesis along with microstructure and macroscale in vivo MR imaging. *Brain Struct Funct.* 2024 Dec;229(9):2339-2367. doi: 10.1007/s00429-024-02838-9. Epub 2024 Aug 5. PMID: 39102068; PMCID: PMC11612014.

Version 1:

Reviewer comments:

Reviewer #1

(Remarks to the Author)

The authors have addressed all my concerns. In particular, verifying the stability of hub distribution in neonates greatly strengthens the solidity of the results.

Reviewer #2

(Remarks to the Author)

The revised version of manuscript is substantially improved and completed according to comments of the reviewers. Analysis of point-by-point comments of both reviewers shows that authors considered carefully every comment and accepted most suggestions. They gave logical and argued explanation where suggestions were not acceptable (title). The text throughout manuscript is revised according to acceptable suggestions of reviewers. Regarding timing and neuroanatomical criteria, the paper now gives more precise descriptions. Additional points in Discussion and revision of some sentences will make understanding of complex network issue easier for non-expert reader. Overall, the manuscript and supplemental material are now in a good shape.

We would like to thank the Reviewers for their positive and constructive comments and the Editor for the opportunity to revise our manuscript. We have provided point-by-point responses below:

Reviewer #1

This study represents a highly novel and instructive investigation characterizing the transcriptomic underpinnings of neonatal brain connectome. Using diffusion MRI data from the dHCP project, the authors identified hub regions of the brain white matter network at birth and further integrated multiple existing brain transcriptomic atlases (uBrain, BrainSpan, Single-cell genomics atlas) to uncover a system and cross-scale molecular blueprint on the origin of cortical hub. They found that the neonatal hub pattern is strongly associated with gene expression in the transient subplate and intermediate layers during mid-gestation, particularly involving genes related to dendritic morphology and axonal guidance, and predominantly expressed by excitatory neurons and astrocytes. The authors are domain experts in neuroimaging-transcriptomic analysis. For connectome analysis, they effectively addressed common pitfalls such as node parcellation, fiber smoothing or threshold selection. For genetic analysis, the uBrain atlas represents as a methodological contribution from their prior work. Overall, this study leverages rare data and rigorous methods to provide novel insights into the developmental timing and genetic regulation of canonical white matter network architecture. It is encouraging to see these findings, and only a few suggestions remain that do not affect the manuscript's suitability for publication.

R1. We thank the Reviewer for their positive comments.

1) The primary challenge in identifying the transcriptomic basis of neonatal hub regions lies in establishing a reliable hub distribution. The authors have conducted solid validations regarding parcellation schemes, network thresholds, and individual similarity. However, I have suggestions for further strengthening.

a) Considering the rapid changes of brain at early stage, is there any development effect of hub location shortly after birth? If such changes are significant, should they be incorporated into the gene-level analyses?

R2. Despite the rapid changes occurring during early brain development, in previous studies, we and others have demonstrated that core network connections are relatively stable across the perinatal period.^{1,2} To test this in the current cohort, we generated time-resolved, group-level networks using all individuals scanned within the same gestational week (i.e., all scanned in the 37th week, 38th week etc), and correlated nodal degree at each week with the full group-averaged network. This confirmed that node degree was highly stable in the weeks after birth, with correlation across gestational weeks ranging from 0.91 (37 weeks GA) to 0.99 (40 weeks GA). As expected, age ranges with more samples produced a stronger correlation to the full group-averaged network (see also **R3** below). We have now included this figure with other validation results in the main text manuscript with a description of the results (**Figure 2**; pg 5).

Figure R1: Convergence of group-level nodal degree over different gestational weeks. A group-averaged network was created using all individual's scanned within a particular gestational age (GA) week (ranging from 9 scans in the 37th week, to 49 scans in the 40th week). The nodal degree profile of each age group was correlated with the nodal degree profile of the group-averaged network constructed from all individuals.

b) the convergence of group-level nodal degree profiles with increasing sample size could be evaluated (is a sample size of 200+ sufficient?);

R3. To assess convergence of nodal degree profiles over increasing sample sizes, we performed a bootstrap analysis sampling random subsets of 1 to N participants (N = 208) with replacement 100 times. For each subsample, we generated a group-level network and correlated its degree distribution with that of the full group. We found that that the degree profiles quickly converged over increasing sample sizes, with a group-level network constructed from 10 individuals achieving a correlation of ~0.97 with the full cohort (see figure below). We have now included this figure with other validation results in the main text manuscript with a description of the results (**Figure 2**; pg 5).

Figure R2: Convergence of group-level nodal degree profiles over increasing sample sizes. We performed a bootstrap analysis selecting a subsample of individual networks of increasing size (repeated 100 times for each sample size) to construct a group-averaged network. Correlation between nodal degree in the bootstrapped sample and the whole group-averaged network is shown for each subsample size. The red line indicates the average across 100 iterations, while the shaded red area indicates the standard deviation.

c) Figure S4 shows consistent associations between nodal degree and gene expression across three network resolutions; however, the reproducibility of specific genes across resolutions is not reported. If the identified gene sets are largely overlapping, reporting this would largely strengthen the robustness; if not, only focusing on the resolution-robust gene subsets may be a good choice.

R4. As the gene-hub association tests were conducted based on node degree averaged within μ Brain atlas parcels and there was a high spatial correlation between nodal degree maps across different resolutions before averaging within parcels, we anticipated strong correlations between gene-level associations across different resolutions. To test this, we calculated the correlation between GLM coefficients for each gene association test using mean degree calculated across the different parcellation resolutions, we found that gene-hub associations were highly robust across parcellation schemes (μ Brain₆₀: μ Brain₉₀ $r = 0.996$; μ Brain₉₀: μ Brain₁₂₀ $r = 0.989$). In total, using the μ Brain₆₀ parcellation, we identified 516 unique genes with significant associations with node degree. Of these, 490 were identified and reported in the main text using the μ Brain₉₀ parcellation (94.9%). Using the μ Brain₁₂₀ parcellation, we identified 342 unique genes, of which 340 were identified using μ Brain₉₀ (99.4%). We have included additional text in the Results section describing these findings (pg 6).

d) some of the validation results in the supplemental figures could be integrated into Figure 1 to better illustrate the stability of hub distribution.

R5. We have now included these results and those derived above (see **R2 & R3**) into a new **Figure 2** to better illustrate the stability of our network results over different parameter settings and analysis scenarios.

1) In the intro part, they mentioned a few times of the modular structure of human brain connectome, which seems unrelated to the main focus. I recommend removing them.

R6. We have now removed this term from the introduction.

2) The sentence “The central positioning of hubs within brain networks supports integration across functional sub-systems; a framework that is supported by experimental tracer studies in other mammalian species” conveys a crucial point. Could the authors elaborate on how these hubs support specific functional subsystems?

R7. We have now included additional references to expand upon this point in the Introduction:

Converging lines of evidence from tracer studies,^{3,4} human neuroimaging^{5–8} and experimental models⁹ suggest that the central positioning of hubs within brain networks provides critical infrastructure to integrate information across functional sub-systems. Human fMRI studies show that network hubs activate in concert with segregated, task specific networks^{6,8,10} and, much like in traffic and information networks, disruption or damage to hub nodes can impart disproportionately widespread disturbances to wider network communication.^{11–17} These observations are supported by experimental tracer studies in other mammalian species.^{4,18–21} and borne out by anatomical lesion studies^{12,22–24} and evidence of network hub alterations across a diverse range of psychiatric, neurodevelopmental and neurological brain disorders.^{22,25–27}

3) The sentence “However, computational simulations predict that disruptions to brain hubs and their connections have far-reaching and adverse effects on overall network function” seems lack a clear contrast with the preceding context. Please make a further clarification.

R8. We have revised this sentence.

4) In addition to multiple comparison correction (FDR), null model corrections (e.g., spin tests, random rewiring) are also critical in gene-hub association analyses and may need to be considered.

R9. This is an important point. We can confirm that in addition to multiple comparison correction with FDR for gene-hub associations, we performed random network re-wiring to establish that network hubs formed a ‘rich-club’ network core in the neonatal connectome.

To further account for potential confounding effects of spatial autocorrelation inflating associations between hub location and gene expression, we have now performed additional analyses. Using the software tool *BrainSMASH*, we generated 1000 surrogate maps with spatial autocorrelation and areal weights matched to the group average node degree maps. For each gene association, we re-fit the statistical model to each surrogate map and calculated a null distribution of coefficients for surrogate gene-hub associations. We found that out of the 653 significant gene associations (from 581 unique genes) originally reported (at $p_{\text{FDR}} < 0.05$), all but 9 associations (from 6 genes) passed an additional significance threshold of $p < 0.05$ based on a null distribution of random spatially autocorrelated associations (p_{spatial}). Although only a small proportion of associations failed this additional statistical test, we repeated all analyses after removing the 6 genes, detecting only minor differences in the reported findings. We have now updated our results in **Tables S2-S5, Figures 3 & 4** and in the main text to account for the additional correction step.

We have included the additional analysis steps in the Methods section (pg. 15).

5) The uBrain atlas only covers up to 21 gestational weeks, while the neonatal brain scans were acquired at a mean age of around 40 weeks. Given the substantial brain changes during this period, particularly the disappearance of the subplate zone implicated in the current findings, it would be helpful to further discuss this temporal gap.

R10. We agree that our observations are limited by the relatively short time window of the microarray data available to us. One of the main aims of this study was to examine how regional patterns of gene expression at mid-gestation are related to formation of network hubs by the time of normal birth. Examination of the concurrent changes in regional gene expression and cortical connectivity over gestation represents a related research question and would require both i) improved spatial and temporal sampling of prenatal gene expression data and ii) reliable and robust fetal diffusion MRI acquisitions. Unfortunately, despite a growing catalogue of comprehensive single-cell RNA-seq studies, including those that span the whole of gestation (e.g.: Herring et al., 2022; Velmeshev et al., 2023), there are no datasets of prenatal gene expression that encompass 20-40 weeks over multiple cortical regions at a spatial resolution comparable to the LMD microarray data used in μ Brain. Similarly, while fetal dMRI acquisition is rapidly improving, significant challenges remain to be addressed in terms of image reconstruction and modelling of the diffusion signal to enable high-resolution, whole-brain tractography. Previously, using the BrainSpan RNA-seq data, we have examined the associations between variation in gene expression between 12 and 37 weeks and cortical morphology in neonates (Ball et al., PLoS Biology, 2020). However, this data is limited to just 11 cortical regions-of-interest, each encompassing tissue from the full cortical anlage (cortical plate, subplate, proliferative zones). In this study, we have used the same data to examine temporal variations in hub gene enrichment, finding that hub+ genes were enriched in both early and late mid-prenatal windows (13 – 24PCW). As such, we provide a snapshot of the process and await new datasets that offers both spatial and temporal resolution during this time period, recognising that this is extremely challenging data to acquire. In the meantime, we have made clear the limitations of these data in the Discussion (pg 11).

6) The neonatal dMRI scan contains multi-shell DWI volumes, however, the authors seems only employed single-shell models for diffusion modeling. Please clarify the rationale.

R11. As in our previous study,²⁸ we modelled the diffusion signal using only the $b=0$ and $b=1000$ MRI shells due to the lower SNR of higher b -values in the neonatal dHCP data. In prior work performed in preparation for this study, we found a single shell approach improved the definition of FODs as compared to multishell approaches, see also Dhollander et al.²⁹

We have amended the text as follows to clarify this (pg 13):

As in our previous work, we modelled the diffusion signal using only the 0 and 1000s/mm² b-value diffusion volumes. This approach has been shown to provide better definition of FODs in neonatal data compared to multi-shell approaches

7) It may be helpful to split Figure 3 into two figures according to the current result sections, which could improve readability.

R12. As we have added an additional figure into the main text (see **R5**) we have chosen to keep **Figure 3** (now **Figure 4**) as a single figure. We have increased white space around panels and revised the legend to improve readability.

Reviewer #2

The manuscript Transcriptomic divergence of network hubs in the prenatal human brain is an ambitious attempt to combine two sets of data (MR and transcriptomic atlasing), two developmental periods (mid-gestation and newborn), in order to get new data on development of the cortical network hubs in the developing prenatal human brain. According to authors, this concept can be called “binary topology”. The subject is very important because data on developing networks during first half of gestation are insufficient, but very important for analysis of major developmental disorders, especially schizophrenia. The abridgements of these two sets of secondary data was performed by careful statistical evaluation of relationship between newborn connectomic data, parcellation of telencephalic pallium and genes expressed at mid-gestation in cortical regions with high network degree at birth. The data are very well documented, illustrated and adequately complemented in supplemental material. The paper offers original conclusions about role of early compartmentalization of transcription factors, claiming that a head start in the formation of the connectivity reduces the high metabolic cost of establishing cortical long-range connections. The data also fill the gap in current neuroimaging studies, where dynamically changing transient anatomical compartments and specific neural and glial classes are not considered. In respect to this last question, they found that excitatory projection neurons in deep layers, such as subplate and layer 6, show specific spatially resolved prenatal gene expression, which are important for establishment of network hubs prior to the time of birth.

R13. We thank the Reviewer for their positive and constructive comments.

Conceptually. I have some difficulties in interpreting tight association between transcriptomic divergence in mid-gestation and MR connectomic data in newborn. First of all, compartments in midgestational brain are actually tangentially oriented from pole to pole and from limbus to basis of the brain. While arealization is event of second half of gestation and I imagine that hubs are focal points of unknown termination and unknown origin of pathways. I suggest that this problem tangential vs hub discuss with somewhat more elaborative approach. The data on development of connectivity are very important for different disciplines and in this respect paper may have significant impact on broader audience.

R14. The Reviewer raises an important discussion on the timing and extent of arealisation in the developing cortex. As noted, early organisational events in the cortex are guided by broad morphogenic concentration gradients along the major spatial axes of the developing brain yet the mature cortex comprises anatomically

and functionally distinct areas. The timing and extent of cortical arealisation during gestation remains a topic of debate.^{30–35} Recent neuroimaging evidence has shown that the geometry of the cortex plays a significant role in the organisation of network connectivity, with structural and functional properties of the mature cortex varying continuously across the cortical sheet, reflecting early spatial gradients.^{36–38} Yet at the cellular level, areal differences in gene expression are evident by 20 weeks of gestation.^{30,39,40} A recent large-scale single-cell transcriptomic study identified both modes of arealisation in the cortical plate at 20PCW, observing gradual transitions along the anterior-posterior axis overlaid by sharp transitions between similar cell types in anatomically adjacent areas.⁴¹ Therefore, while the positioning of early network hubs is likely to reflect locales at the confluence of broader organisational gradients, it does not discount the possibility that hubs boundaries are defined early as targets for axonal outgrowth. As noted above in **R10**, our observations are limited by the data currently available to us, we look forward to being able to examine these important questions with higher density tissue sampling and improved acquisition protocols in the future.

We have now added some points on this to the Discussion (pg 10).

Title: The title does not give indication on extensive newborn MR data. Maybe this can be added in title. Description of Figure 2 can serve as a model.

R15. We thank the Reviewer for this suggestion. We would prefer to keep the title as is but have edited the abstract to highlight that the study is based on high resolution dMRI from 208 neonates.

Abstract. Row 41. Transmodal is also association cortex.

R16. We agree with this statement and have revised the text accordingly.

Row 47. Prior to time of birth. It is not precise enough. Can you mention significance for study of neurodevelopmental disorders?

R17. We have amended this sentence. We have also added a sentence to highlight relevance of our findings to neurodevelopmental disorders.

Introduction. Row 69-72. Expression that anatomical lesions correlate with hubs is too strong, but this is a good idea to do in the future.

R18. See also **R7**, we have expanded this section of the Introduction to provide more information on the links between hubs, system connectivity and the potential widespread impact of disturbances to hub nodes, including lesions.

Row 85. By 30 weeks of gestation pathways all wiring is in position. This is actually the last trimester.

R19. We have amended this sentence.

Row 76-79. Excellent compression of current evidence, but maybe difficult for non-specialist to reveal these complex relationships.

R20. We agree that this is a complex topic and, as discussed above in **R14**, one that is not yet resolved. We hope that the additional Discussion points now added to the text (**R14**) provide the additional background information for the interested reader.

Row 84-94. Despite advances of MR analysis of connectome there is no agreement how many parcels we have. For example, classical neuroanatomist Rose described almost hundred areas in entorhinal cortex.

R21. We agree and because of this, chose to examine network properties over three different parcellation resolutions composed of between 528 to 1032 subparcels. We found that our results were robust to different parcellation schemes. Future research might focus on using alternative structural parcellations, or as noted in **R25**, incorporating multimodal data e.g.: fMRI to define functional nodes as well.

Results. Row 134. Be more precise, across radial or tangential plane?

R22. See also **R14**. We have added text to clarify that in this study, we were interested in testing associations across a tangential plane, examining spatial variation in gene expression independently within each layer. In previous studies, we and others have examined radial differences in gene expression, i.e.: *across* tissue compartments, in this dataset.^{39,42}

Row 137. Add postmigratory compartments. Mention that marginal zone cannot be visualized and it is difficult to perform transcriptomic analysis.

R23. As noted by the Reviewer, while some transcriptomic data is available in the original resource, the uBrain digital atlas does not contain labels for the marginal and subpial zones due to their small size, difficulties encountered during image reconstruction and relatively poor sampling compared to other compartments. We have included these points in the limitations section of the Discussion (pg 12):

Similarly, due to difficulties in image reconstruction, the μ Brain atlas doesn't contain labels for the outermost cortical layers: the marginal and subpial zones, thus our analysis does not consider the potential role in the development of cortical connections of early-born cell populations in these zones.

Row 234. Because of these results few words on neurodevelopmental disorders should be included in Abstract.

R24. We have now edited the abstract to highlight relevance to neurodevelopmental disorders. See also **R17**.

Discussion. I would emphasize significance of binary and multimodal approach for association between hubs and transcriptomics of transient compartments.

R25. We have added a statement in the Discussion highlighting the future added value of new multimodal approaches for understanding connectivity in the developing brain (pg 12):

Incorporation of multimodal data, in the form of morphometric networks, or structural-functional associations will likely provide additional insight into the development of brain connectivity in future studies.

It is important to elaborate so-called limitations of the study from Row 309 to 320. Please consider prominent changes which occur between mid-gestation and newborn age. For example, associative neurons of layer 3, which send long connections are not even born before 13 weeks post conception.

R26. See also **R10.** We had added additional statements to the Discussion acknowledging that our observations are limited by the relatively short time window of the microarray data currently available for analysis

Row 287. When quoting about origin of subplate consider that only papers by Rakic and Kostovic group give data on origin of subplate in human and monkey (see A. Duque, Z. Krsnik, I. Kostović, & P. Rakic, Secondary expansion of the transient subplate zone in the developing cerebrum of human and nonhuman primates, Proc. Natl. Acad. Sci. U.S.A. 113 (35) 9892-9897, <https://doi.org/10.1073/pnas.1610078113>(2016).

R27. Thank you for highlighting this study, we have made sure to include it when referencing the development of the human subplate (pg. 11)

References, I recommend two references: one about multimodal gradients of cortical organization and the other, because of frequent mentioning of synapses, recent paper on multiscale analysis of synaptic distribution:

Wang Y et al. Multimodal gradients unify local and global cortical organization. Nat Commun. 2025 Apr 25;16(1):3911. doi: 10.1038/s41467-025-59177-4. PMID: 40280959; PMCID: PMC12032020.

Kostović I. Development of the basic architecture of neocortical circuitry in the human fetus as revealed by the coupling spatiotemporal pattern of synaptogenesis along with microstructure and macroscale in vivo MR imaging. Brain Struct Funct. 2024 Dec;229(9):2339-2367. doi: 10.1007/s00429-024-02838-9. Epub 2024 Aug 5. PMID: 39102068; PMCID: PMC11612014.

R29. Thank you for bringing these interesting papers to our attention, we have now included them in our Discussion at the appropriate locations (pg 11, pg 12). See also **R25.**

Additional references

1. Ball, G. *et al.* Rich-club organization of the newborn human brain. *Proc. Natl. Acad. Sci.* **111**, 7456–7461 (2014).
2. Batalle, D. *et al.* Early development of structural networks and the impact of prematurity on brain connectivity. *Neuroimage* **149**, 379–392 (2017).
3. Coletta, L. *et al.* Network structure of the mouse brain connectome with voxel resolution. *Sci. Adv.* **6**, eabb7187 (2020).
4. Zamora-López, G., Zhou, C. & Kurths, J. Cortical hubs form a module for multisensory integration on top of the hierarchy of cortical networks. *Front. Neuroinformatics* **4**, 1 (2010).
5. Heuvel, M. P. van den & Sporns, O. An Anatomical Substrate for Integration among Functional Networks in Human Cortex. *J. Neurosci.* **33**, 14489–14500 (2013).
6. Gordon, E. M. *et al.* Three Distinct Sets of Connector Hubs Integrate Human Brain Function. *Cell Rep.* **24**, 1687-1695.e4 (2018).
7. Bertolero, M. A., Yeo, B. T. T. & D’Esposito, M. The modular and integrative functional architecture of the human brain. *Proc. Natl. Acad. Sci. U. S. A.* **112**, E6798–E6807 (2015).
8. Cole, M. W. *et al.* Multi-task connectivity reveals flexible hubs for adaptive task control. *Nat. Neurosci.* **16**, 1348–1355 (2013).
9. Towilson, E. K., Vértes, P. E., Ahnert, S. E., Schafer, W. R. & Bullmore, E. T. The Rich Club of the *C. elegans* Neuronal Connectome. *J. Neurosci.* **33**, 6380–6387 (2013).
10. Bertolero, M. A., Yeo, B. T. T. & D’Esposito, M. The modular and integrative functional architecture of the human brain. *Proc. Natl. Acad. Sci. U. S. A.* **112**, E6798–E6807 (2015).
11. de Reus, M. A. & van den Heuvel, M. P. Simulated rich club lesioning in brain networks: a scaffold for communication and integration? *Front. Hum. Neurosci.* **8**, (2014).
12. Gratton, C., Nomura, E. M., Pérez, F. & D’Esposito, M. Focal brain lesions to critical locations cause widespread disruption of the modular organization of the brain. *J. Cogn. Neurosci.* **24**, 1275–1285 (2012).
13. Lynch, C. J. *et al.* Precision Inhibitory Stimulation of Individual-Specific Cortical Hubs Disrupts Information Processing in Humans. *Cereb. Cortex N. Y. N 1991* **29**, 3912–3921 (2019).
14. Vetere, G. *et al.* Chemogenetic Interrogation of a Brain-wide Fear Memory Network in Mice. *Neuron* **94**, 363-374.e4 (2017).
15. Alstott, J., Breakspear, M., Hagmann, P., Cammoun, L. & Sporns, O. Modeling the impact of lesions in the human brain. *PLoS Comput. Biol.* **5**, e1000408 (2009).
16. Albert, R., Jeong, H. & Barabasi, A. L. Error and attack tolerance of complex networks. *Nature* **406**, 378–382 (2000).
17. Kaiser, M., Martin, R., Andras, P. & Young, M. P. Simulation of robustness against lesions of cortical networks. *Eur. J. Neurosci.* **25**, 3185–3192 (2007).
18. Markov, N. T. *et al.* Cortical High-Density Counterstream Architectures. *Science* **342**, 1238406 (2013).
19. Bertolero, M. A., Yeo, B. T. T., Bassett, D. S. & D’Esposito, M. A mechanistic model of connector hubs, modularity and cognition. *Nat. Hum. Behav.* **2**, 765–777 (2018).
20. Gómez-Gardeñes, J., Zamora-López, G., Moreno, Y. & Arenas, A. From Modular to Centralized Organization of Synchronization in Functional Areas of the Cat Cerebral Cortex. *PLoS ONE* **5**, e12313 (2010).
21. Markov, N. T. *et al.* The role of long-range connections on the specificity of the macaque interareal cortical network. *Proc. Natl. Acad. Sci. U. S. A.* **110**, 5187–5192 (2013).
22. Crossley, N. A. *et al.* The hubs of the human connectome are generally implicated in the anatomy of brain disorders. *Brain* **137**, 2382–2395 (2014).
23. Reber, J. *et al.* Cognitive impairment after focal brain lesions is better predicted by damage to structural than functional network hubs. *Proc. Natl. Acad. Sci.* **118**, e2018784118 (2021).
24. Warren, D. E. *et al.* Network measures predict neuropsychological outcome after brain injury. *Proc. Natl. Acad. Sci.* **111**, 14247–14252 (2014).
25. Sporns, O. Towards network substrates of brain disorders. *Brain J. Neurol.* **137**, 2117–2118 (2014).
26. van den Heuvel, M. P. & Sporns, O. A cross-disorder connectome landscape of brain dysconnectivity. *Nat. Rev. Neurosci.* **20**, 435–446 (2019).
27. Rubinov, M. & Bullmore, E. Schizophrenia and abnormal brain network hubs. *Dialogues Clin. Neurosci.* **15**, 339 (2013).

28. Oldham, S., Mansour L., S. & Ball, G. Perinatal development of structural thalamocortical connectivity. *Imaging Neurosci.* **3**, imag_a_00418 (2025).
29. Dhollander, T., Mito, R. & Connelly, A. Single-Shell 3-Tissue CSD (SS3T-CSD) modelling of developing HCP (dHCP) diffusion MRI data. in *Organisation for Human Brain Mapping* (Rome, Italy, 2019).
30. Bhaduri, A. *et al.* An atlas of cortical arealization identifies dynamic molecular signatures. *Nature* **598**, 200–204 (2021).
31. Cadwell, C. R., Bhaduri, A., Mostajo-Radji, M. A., Keefe, M. G. & Nowakowski, T. J. Development and Arealization of the Cerebral Cortex. *Neuron* **103**, 980–1004 (2019).
32. Huntenburg, J. M., Bazin, P.-L. & Margulies, D. S. Large-Scale Gradients in Human Cortical Organization. *Trends Cogn. Sci.* **22**, 21–31 (2018).
33. Hilgetag, C. C. & Goulas, A. 'Hierarchy' in the organization of brain networks. *Philos. Trans. R. Soc. B Biol. Sci.* **375**, 20190319 (2020).
34. Elsen, G. E. *et al.* The protomap is propagated to cortical plate neurons through an Eomes-dependent intermediate map. *Proc. Natl. Acad. Sci.* **110**, 4081–4086 (2013).
35. O'Leary, D. D. M., Chou, S.-J. & Sahara, S. Area Patterning of the Mammalian Cortex. *Neuron* **56**, 252–269 (2007).
36. Pang, J. C. *et al.* Geometric constraints on human brain function. *Nature* **618**, 566–574 (2023).
37. Hansen, J. Y. *et al.* Mapping gene transcription and neurocognition across human neocortex. *Nat. Hum. Behav.* **5**, 1240–1250 (2021).
38. Vogel, J. W. *et al.* Deciphering the functional specialization of whole-brain spatiomolecular gradients in the adult brain. *Proc. Natl. Acad. Sci.* **121**, e2219137121 (2024).
39. Miller, J. A. *et al.* Transcriptional landscape of the prenatal human brain. *Nature* **508**, 199–206 (2014).
40. Li, M. *et al.* Integrative functional genomic analysis of human brain development and neuropsychiatric risks. *Science* **362**, eaat7615 (2018).
41. Qian, X. *et al.* Spatial transcriptomics reveals human cortical layer and area specification. *Nature* **644**, 153–163 (2025).
42. Ball, G. *et al.* Molecular signatures of cortical expansion in the human foetal brain. *Nat. Commun.* **15**, 9685 (2024).